# LCM: Locally Constrained Compact Point Cloud Model for Masked Point Modeling

**Yaohua Zha**[1,2]   **Naiqi Li**[1]   **Yanzi Wang**[1]   **Tao Dai**[3] *
**Hang Guo**[1]   **Bin Chen**[4]   **Zhi Wang**[1]   **Zhihao Ouyang**[5]   **Shu-Tao Xia**[1,2]

[1]Tsinghua Shenzhen International Graduate School, Tsinghua University
[2]Institute of Visual Intelligence, Pengcheng Laboratory
[3]College of Computer Science and Software Engineering, Shenzhen University
[4]Harbin Institute of Technology, Shenzhen    [5]Bytedance Inc.
zyh1614399882@gmail.com

## Abstract

The pre-trained point cloud model based on Masked Point Modeling (MPM) has ex-hibited substantial improvements across various tasks. However, these models heav-ily rely on the Transformer, leading to quadratic complexity and limited decoder, hindering their practice application. To address this limitation, we first conduct a comprehensive analysis of existing Transformer-based MPM, emphasizing the idea that redundancy reduction is crucial for point cloud analysis. To this end, we propose a **L**ocally constrained **C**ompact point cloud **M**odel (LCM) consisting of a lo-cally constrained compact encoder and a locally constrained Mamba-based decoder. Our encoder replaces self-attention with our local aggregation layers to achieve an elegant balance between performance and efficiency. Considering the varying information density between masked and unmasked patches in the decoder inputs of MPM, we introduce a locally constrained Mamba-based decoder. This decoder ensures linear complexity while maximizing the perception of point cloud geometry information from unmasked patches with higher information density. Extensive experimental results show that our compact model significantly surpasses existing Transformer-based models in both performance and efficiency, especially our LCM-based Point-MAE model, compared to the Transformer-based model, achieved an improvement of 1.84%, 0.67%, and 0.60% in average accuracy on the three variants of ScanObjectNN while reducing parameters by **88%** and computation by **73%**. Code is available at https://github.com/zyh16143998882/LCM.

## 1   Introduction

3D point cloud perception, as a crucial application of deep learning, has achieved significant success across various areas such as autonomous driving, robotics, and virtual reality. Recently, point cloud self-supervised learning [1, 58, 60], capable of learning universal representations from extensive unlabeled point cloud data, has gained much attention. Among which, masked point modeling (MPM) [8, 37, 60, 62, 65, 66], as an important self-supervised paradigm, has become mainstream in point cloud analysis and has gained immense success across diverse point cloud tasks.

The classical MPM [37, 60, 65], inspired by masked image modeling [2, 22, 59] (MIM), divides point clouds into patches and uses a standard Transformer [46] backbone. It randomly masks some patches in the encoder input and combines the unmasked patch tokens with randomly initialized

---

*Corresponding author. ✉ daitao.edu@gmail.com

masked patch tokens in the decoder input. It predicts the geometric coordinates or semantic features of the masked patches from the decoder output tokens, enabling the model to learn universal 3D representations. Despite the significant success, two inherent issues of Transformers still limit their practical deployment.

The first issue is that the Transformer architecture leads to quadratic complexity and huge model sizes. As shown in Figure 1 (a) and (b), MPM methods like Point-MAE [37] based on standard Transformer [46] require 22.1M parameters and complexity exponentially grows with an increase in the length of input patches. However, in practical point cloud applications, models are often deployed on embedded devices such as robots or VR headsets, where strict constraints exist regarding the model's size and complexity. In this context, lightweight networks such as PointNet++ [40] are more popular in practical applications due to their lower parameter requirement (only 1.5M) even though they may have inferior performance.

Another issue is that when Transformers [46] are used as decoders in Masked Point Modeling (MPM), their potential to reconstruct masked patches with lower information density is limited. In the decoder input of MPM, randomly initialized masked tokens with lower information density are typically concatenated with unmasked tokens with higher information density and fed into the Transformer-based decoder. The self-attention layers then learn to process these tokens of varying information density based on loss constraints. However, relying solely on the loss to learn this objective is challenging due to the lack of explicit importance guidance for different densities. Additionally, in Section 5.1, we further explain from an information theory perspective that the self-attention mechanism, as a higher-order processing function, can limit the model's reconstruction potential.

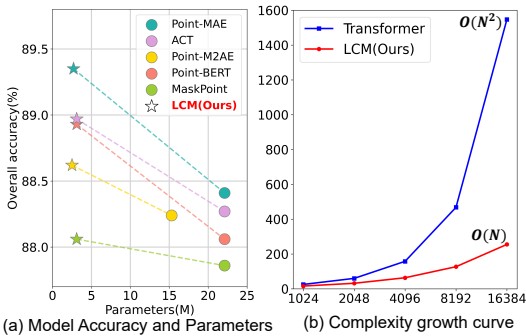

(a) Model Accuracy and Parameters  (b) Complexity growth curve

Figure 1: Comparison of our LCM and Transformer in terms of performance and efficiency.

To address the above issues, as shown in Figure 2, we first conducted a comprehensive analysis of the effects of different top-K attention on the performance of the Transformer model, emphasizing the idea that redundancy reduction is crucial for point cloud analysis. To this end, we propose a **L**ocally constrained **C**ompact point cloud **M**odel (LCM), consisting of a locally constrained compact encoder and a locally constrained Mamba-based decoder, to replace the standard Transformer. Specifically, based on the idea of redundancy reduction, our compact encoder replaces self-attention with our local aggregation layers to achieve an elegant balance between performance and efficiency. The local aggregation layer leverages static local geometric constraints to aggregate the most relevant information for each patch token. Since static local geometric constraints only need to be computed once at the beginning and are shared across all layers, it avoids dynamic attention computations in each layer, significantly reducing complexity. Furthermore, it uses only two MLPs for information mapping, greatly reducing the network's parameters.

In our decoder design, considering the varying information density between masked and unmasked patches in the inputs of MPM, our decoder introduces the State Space Model (SSM) from Mamba [13, 16, 19, 30, 71] to replace self-attention, ensuring linear complexity while maximizing the perception of point cloud geometry information from unmasked patches with higher information density. However, as discussed in Section 5.4, the directly replaced SSM layer exhibits a strong dependence on the order of input patches. Inspired by our compact encoder, we migrate the idea of local constraints to the feedforward neural network of our Mamba-based decoder, proposing the Local Constraints Feedforward Network (LCFFN). This eliminates the need to explicitly consider the sequence order of input in SSM layers because the subsequent LCFFN can adaptively exchange information among geometrically adjacent patches based on their implicit geometric order.

Our LCM is a universal point cloud architecture designed based on the characteristics of the point cloud to replace the standard Transformer. It can be trained from scratch or integrated into any existing pretraining strategy to achieve an elegant balance between performance and efficiency. For example, the LCM model pre-trained based on the Point-MAE strategy requires only 2.7M parameters, which is about **10 ×** efficient compared to the original Transformer with 22.1M. Furthermore, in

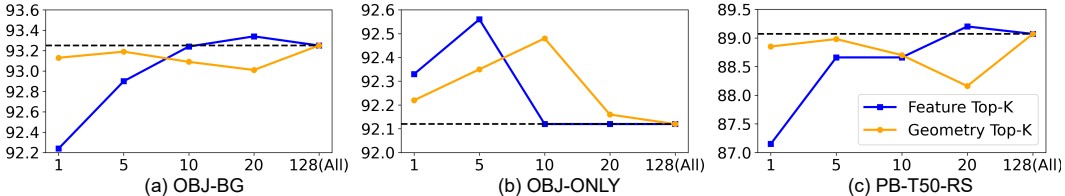

Figure 2: The effect of using top-K attention in feature space and geometric space by the Transformer on the classification performance in ScanObjectNN, all results are the averages of ten repeated experiments.

terms of performance, compared to the Transformer, the LCM shows significant improvements of 1.84%, 0.67%, and 0.60% in average classification accuracy of three variants of ScanObjectNN [44]. Additionally, in the detection task of ScanNetV2 [6], there are also significant improvements of **+5.2%** on $AP_{25}$ and **+6.0%** on $AP_{50}$.

We summarize the contributions of our paper as follows: **1)** We propose a locally constrained compact encoder, which leverages static local geometric constraints to aggregate the most relevant information for each patch token, achieving an elegant balance between performance and efficiency. **2)** We propose a locally constrained Mamba-based decoder for masked point modeling, which replaces the self-attention layer with Mamba's SSM layer and introduces a locally constrained feedforward neural network to eliminate the explicit dependency of Mamba on the input sequence order. **3)** Our locally constrained compact encoder and locally constrained Mamba-based decoder together constitute the efficient backbone LCM for masked point modeling. We combine LCM with various pretraining strategies to pre-train efficient models and validate our model's superiority in efficiency and performance across various downstream tasks.

## 2    Related Work

**Point Cloud Self-supervised Pre-training.** Point cloud self-supervised pre-training [47, 54, 55, 58, 60] has achieved remarkable improvement in many point cloud tasks. This approach first applies a pretext task to learn the latent 3D representation and then transfers it to various downstream tasks. PointContrast [58] and CrossPoint [1] initially explored utilizing contrastive learning [36, 43] for learning 3D representations, which achieved some success; however, there were still some shortcomings in capturing fine-grained semantic representations. Recently, masked point modeling methods [37, 60–62] demonstrated significant improvements in learning fine-grained point cloud representations through masking and reconstruction. Many methods [4, 8, 21, 41, 66] have attempted to leverage multimodal knowledge to assist MPM in learning more generalized representations, yielding significant improvements. After obtaining a pre-trained point cloud model, many works [18, 64, 67, 70] remain to explore parameter-efficient fine-tuning methods to better adapt these pretrained models to a variety of downstream tasks. While the pre-trained models mentioned above have achieved tremendous success, they all rely on the Transformer architecture. In this paper, we focus on designing a more efficient architecture to replace the Transformer in these methods, significantly reducing computational and resource requirements.

## 3    Methodology

### 3.1    Observation of Top-K Attention

Standard Transformer [46] architecture requires computing the correlation between each patch with all input patches, resulting in quadratic complexity. While this architecture performs well in language data, its effectiveness in point cloud data has been underexplored. Not all points are equally important. As illustrated in Figure 3, the key points for aircraft recognition are mainly distributed on the wings, while for vase recognition, they are primarily located on the bottom of the vase. Therefore, directly skipping the attention computation for less important points provides a straightforward solution.

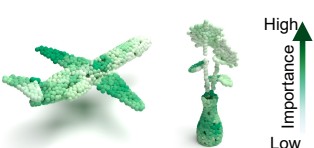

Figure 3: Point heatmap.

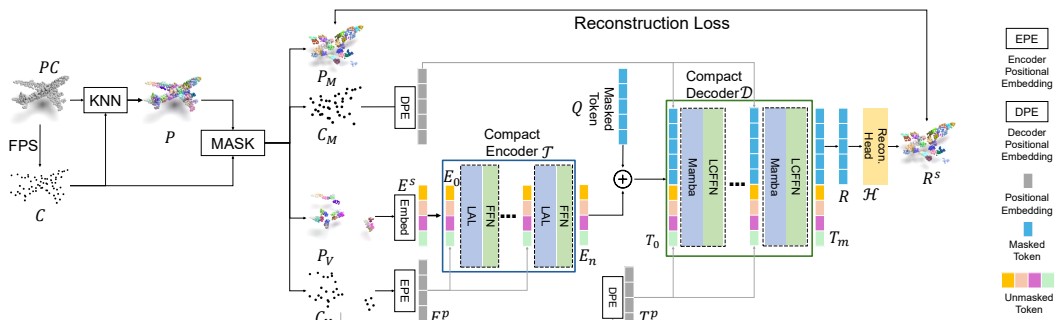

Figure 4: The pipeline of our Locally Constrained Compact Model (LCM) with Point-MAE pre-training. Our LCM consists of a locally constrained compact encoder and a locally constrained Mamba-based decoder.

We first replaced the computation of global attention for all patch tokens with calculations top-K attentions in both feature and geometric space. As shown in Figure 2, our empirical observations indicate that: **1)** In self-attention, it is often more effective to use attention weights based on the top-K most important patch tokens rather than using all patch; **2)** Compared to using top-K attention in a dynamic feature space, employing top-K attention in a static geometric space yields nearly identical representational capacity and offers the advantage of a smaller K value. Although this naive method of masking out unimportant attention still exhibits quadratic complexity, this redundancy reduction idea not only brings performance improvements but also provides a direction for further optimizing computational efficiency.

## 3.2 The Pipeline of Masking Point Modeling with LCM

The overall architecture of our Locally constrained Compact Model (LCM) is shown in Figure 4. The specific process is as follows.

**Patching, Masking, and Embedding.** Given an input point cloud $PC \in \mathbb{R}^{L \times 3}$ with $L$ points, we initially downsample a central point cloud $C \in \mathbb{R}^{N \times 3}$ with $N$ points by farthest point sampling (FPS). Then, we perform K-Nearest Neighborhood (KNN) around $C$ to get point patches $P \in \mathbb{R}^{N \times K \times 3}$. Following this, we randomly mask a portion of $C$ and $P$, resulting in masked elements $C_M \in \mathbb{R}^{(1-r)N \times 3}$ and $P_M \in \mathbb{R}^{(1-r)N \times K \times 3}$ and unmasked elements $C_V \in \mathbb{R}^{rN \times 3}$ and $P_V \in \mathbb{R}^{rN \times K \times 3}$, where $r$ denotes the unmask ratio. Finally, we use MLP-based embedding layer (Embed) and position encoding layer (PE) respectively to extract semantic tokens $E_0 \in \mathbb{R}^{rN \times d}$ and central position embedding $E_p \in \mathbb{R}^{rN \times d}$ for the unmasked patches, where $d$ is the feature dimension.

**Encoder.** We employ our locally constrained compact encoder $\mathcal{T}$ to extract features from the unmasked features $E_0$. It consists of $n$ stacked encoder layers, each layer incorporating a local aggregation layer and a feedforward neural network, detailed in Figure 4. For the input feature $E_{i-1}$ of the $i$-th layer, after adding its positional embedding $E^p$, it feeds to the $i$-th encoding layer $\mathcal{T}_i$ to obtain the feature $E_i$. Therefore, the forward process of each encoder layer is defined as:

$$E_i = \mathcal{T}_i(E_{i-1} + E^p), \quad i = 1, ..., n \tag{1}$$

**Decoder.** In the decoding phase, although various MPMs have different decoding strategies, they can generally be divided into feature-level or coordinate-level reconstruction, and their decoders mostly rely on the Transformer architecture. Here, we illustrate the decoding process of our locally constrained Mamba-based decoder using the coordinate-level reconstruction method Point-MAE [37] as an example.

We first concatenate unmasked tokens $E_n \in \mathbb{R}^{rN \times d}$ before the randomly initialized masked tokens $Q \in \mathbb{R}^{(1-r)N \times d}$ to obtain the input $T_0 \in \mathbb{R}^{N \times d}$ for the decoder. Then, we separately calculate the positional encoding for unmasked patches $T_V^p \in \mathbb{R}^{rN \times d}$ and masked patches $T_M^p \in \mathbb{R}^{(1-r)N \times d}$, and then concatenate them together to obtain the positional embeddings $T^p \in \mathbb{R}^{N \times d}$, shared by all layers of the decoder. Finally, for the input feature $T_{i-1}$ of the $i$-th decoder layer, after adding their positional embeddings $T^p$, they are passed into the $i$-th decoder layer $\mathcal{D}_i$ to compute the output

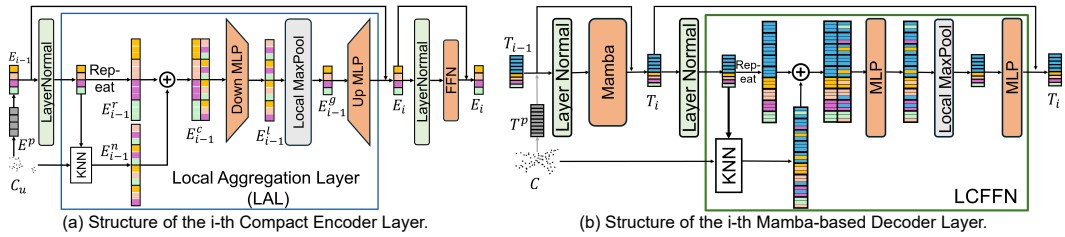

(a) Structure of the i-th Compact Encoder Layer.    (b) Structure of the i-th Mamba-based Decoder Layer.

Figure 5: The structure of $i$-th locally constrained compact encoder layer (a) and $i$-th locally constrained Mamba-based decoder layer (b).

features $\boldsymbol{T_i}$. Therefore, the forward process of each decoder layer is defined as:

$$\boldsymbol{T_i} = \mathcal{D}_i(\boldsymbol{T_{i-1}} + \boldsymbol{T^P}), \quad i = 1, ..., m, \tag{2}$$

**Reconstruction.** We utilize the features $\boldsymbol{R} = \boldsymbol{T_m}[rN :]$ decoded by the decoder to perform the 3D reconstruction. We employ multi-layer MLPs to construct coordinates reconstruction head $\mathcal{H}$ and our reconstruction target is to recover the relative coordinates $\boldsymbol{R}_M = \mathcal{H}(\boldsymbol{R})$ of the masked patches. We use the $\boldsymbol{l_2}$ Chamfer Distance [9] ($\mathcal{CD}$) as reconstruction loss. Therefore, our loss function $\mathcal{L}$ is as follows

$$\mathcal{L} = \mathcal{CD}(\boldsymbol{R_M}, \boldsymbol{P_M}) \tag{3}$$

### 3.3 Locally Constrained Compact Encoder

The classical Transformer [46] relies on the self-attention mechanism to perceive long-range correlations among all patches globally and has achieved great success in language and image domains. However, there remains uncertainty about whether directly transferring a Transformer-based encoder is suitable for point cloud data. Firstly, applications of point clouds are more inclined towards practical embedded devices such as robots or VR headsets. The hardware resources of these devices are limited, imposing higher limits on the model size and complexity, and the Transformer-based backbone demands significantly more resources than traditional networks, as illustrated in Table 1. Secondly, extensive research [34, 40, 50] and our empirical observation as illustrated in Figure 2 also indicate that the perception of local geometry in point cloud data far outweighs the need for global perception. Therefore, the computation of long-range correlations in self-attention leads to a considerable amount of redundant calculations. To address these practical issues, we propose a locally constrained compact encoder.

Our compact encoder consists of $n$ stacked compact encoder layers, each layer comprising a local aggregation layer (LAL) and a feed-forward network (FFN), as shown in Figure 5 (a). For the $i$-th encoder layer, the output ($\boldsymbol{E_{i-1}}$) of the preceding layer, added with the positional embedding and normalized by layer normal, is initially fed to the Local Aggregation Layer (LAL) for aggregating local geometric. Afterward, the result is added to the input residual, passed through layer normalization, and finally fed into a Feed-forward Network (FFN) to obtain the ultimate output feature ($\boldsymbol{E_i}$). This process can be formalized as follows,

$$\boldsymbol{E_i} = \boldsymbol{E_{i-1}} + l_i(n_i^1(\boldsymbol{E_{i-1}}), \boldsymbol{C_u}) \tag{4}$$

$$\boldsymbol{E_i} = \boldsymbol{E_i} + f_i(n_i^2(\boldsymbol{E_i})) \tag{5}$$

where $l_i(\cdot)$ represents the LAM, $n_i^1(\cdot)$ and $n_i^2(\cdot)$ represents layer normalization, and $f_i(\cdot)$ represents the FFN.

In the local aggregation layer, we first use the k-nearest neighbors algorithm based on the central coordinates $\boldsymbol{C_u}$ of the features $\boldsymbol{E_{i-1}}$ to find the $k$ nearest neighbors feature $\boldsymbol{E_{i-1}^n} \in \mathbb{R}^{rkN \times d}$ for each token in $\boldsymbol{E_{i-1}}$. We then replicate each token of $\boldsymbol{E_{i-1}}$ $k$ times and concatenate them with their corresponding neighbors to obtain $\boldsymbol{E_{i-1}^c} \in \mathbb{R}^{rkN \times 2d}$. Next, Down MLP performs a non-linear mapping on all local neighboring features to capture local geometric information. Subsequently, local max pooling is applied to aggregate all local features for each patch. Finally, Up MLP maps all patches to obtain locally enhanced features $\boldsymbol{E_i} \in \mathbb{R}^{rN \times d}$. Our LAL consists of only two simple MLP layers,

significantly reducing the network's parameters. Additionally, since static local geometric constraints only need to be computed once at the beginning and are shared across all layers thereafter, it avoids dynamic attention computations in each layer, significantly reducing computational requirements. It uses only two MLPs for information mapping, greatly reducing the network's parameters.

### 3.4 Locally Constrained Mamba-based Decoder

The decoder for mask point modeling needs to recover information about masked patches based on the features $E_n$ extracted from unmasked patches by the encoder. A common approach is to concatenate the features $E_n$ of unmasked patches before randomly initialized features $Q$ of masked patches as the input to the decoder, as shown in Figure 4. However, at this point, there is a significant difference in information density between features $E_n$ and $Q$. The Transformer architecture and our local aggregation layer both treat each token in the input as equally important initially, it works well when the information density of all tokens is similar. It does not adapt well to cases where there is a large difference in information density in the input.

To efficiently extract more geometric priors from unmasked features $E_n$, we were inspired by the Mamba [13] model in time sequence and proposed using a Mamba-based decoder. This decoder can extract more prior information from the preceding tokens in the sequence based on the input order to aid the learning of subsequent tokens. Initially, we simply replaced the self-attention layer in the original Transformer-based [46] decoder with the state space model (SSM) layer from Mamba. We also sorted the input sequence based on the order of each patch's center point coordinates, creating a naive Mamba-based decoder. Our experiments in Section 8 revealed that although this naive decoder is efficient enough, the simple sorting method cannot effectively model the complex spatial geometry of point clouds and leads to a strong dependence on the order of input patches.

To ensure that the SSM fully perceives the spatial geometry of point clouds, we further introduced the concept of local constraints from the local aggregation layer into the feedforward neural network layer of our decoder, getting the Local Constraints Feedforward Network (LCFFN). By feeding the tokens outputted by the SSM layer into the LCFFN, the LCFFN can implicitly exchange information between geometrically adjacent patches based on their central coordinates. This eliminates the limitation in the SSM layer where explicit sequential input fails to perceive complex geometry fully. Finally, in Section 5.1, we also qualitatively explain from an information theory perspective that this Mamba-based architecture has greater reconstruction potential compared to the Transformer.

Our Mamba-based decoder consists of $m$ stacked decoder layers, each layer comprising a Mamba SSM layer and a local constraints feedforward network (LCFFN), as shown in Figure 5 (b). For the $i$-th decoder layer, we first add the output ($T_{i-1}$) of the previous layer with the positional embeddings ($T^p$) and normalize it through layer normalization. Then, we use the Mamba SSM layer ($s_i(\cdot)$) to perceive geometry from unmasked features and predict masked features. Finally, in the LCFFN ($f_i^l(\cdot)$), we further perceive shape priors based on the central coordinates of each token from its geometrically adjacent tokens. This process can be formalized as follows:

$$T_i = T_{i-1} + s_i(n_i^1(T_{i-1})) \tag{6}$$

$$T_i = T_i + f_i^l(n_i^2(T_i), C) \tag{7}$$

## 4 Experiments

### 4.1 Pre-training

We pre-training our LCM using five different pretraining strategies: Point-BERT [60], MaskPoint [28], Point-MAE [37], Point-M2AE [65], and ACT [8]. For a fire comparison, we use ShapeNet [3] as our pre-training dataset, encompassing over 50,000 distinct 3D models spanning 55 prevalent object categories. For the hyperparameter settings during the pretraining phase, we used the same settings as previous methods.

### 4.2 Fine-tuning on Downstream Tasks

We assess the performance of our LCM by fine-tuning our models on various downstream tasks, including object classification, scene-level detection, and part segmentation.

Table 1: Classification accuracy on real-scanned point clouds (ScanObjectNN). We report the overall accuracy (%) on three variants. "#Params" represents the model's parameters and FLOPs refer to the model's floating point operations. GPT, CL, and MPM respectively refer to pre-training strategies based on autoregression, contrastive learning, and masked point modeling. ○ is the reported results from the original paper. • is the result reproduced in our downstream settings.

| Method | Pretrain | #Params(M) | FLOPs(G) | ScanObjectNN | | |
| --- | --- | --- | --- | --- | --- | --- |
| | | | | OBJ-BG | OBJ-ONLY | PB-T50-RS |
| *Supervised Learning Only* | | | | | | |
| ○ PointNe [39] | ✗ | 3.5 | 0.5 | 73.3 | 79.2 | 68.0 |
| ○ PointNet++ [40] | ✗ | 1.5 | 1.7 | 82.3 | 84.3 | 77.9 |
| ○ PointMLP [32] | ✗ | 12.6 | 31.4 | - | - | 85.2 |
| ○ Transformer [46] | ✗ | 22.1 | 4.8 | 86.75 | 86.92 | 80.78 |
| ○ PointMamba [27] | ✗ | 12.3 | - | 88.30 | 87.78 | 82.48 |
| ○ SFR [63] | ✗ | - | - | - | - | 87.80 |
| • Transformer [46] | ✗ | 22.1 | 4.8 | 91.95 | 91.39 | 86.65 |
| • LCM (Ours) | ✗ | 2.7(↓ 88%) | 1.3(↓ 73%) | 92.77(↑ 0.82) | 91.54(↑ 0.15) | 87.75(↑ 1.10) |
| *Self-Supervised Learning* | | | | | | |
| ○ Point-BERT [60] | MPM | 22.1 | 4.5 | 87.43 | 88.12 | 83.07 |
| ○ MaskPoint [28] | MPM | 22.1 | 4.5 | 89.30 | 88.10 | 84.30 |
| ○ Point-MAE [37] | MPM | 22.1 | 4.8 | 90.02 | 88.29 | 85.18 |
| ○ Point-MAE w/ IDPT [64] | MPM | 23.3 | 7.1 | 91.22 | 90.02 | 84.94 |
| ○ Point-MAE w/ DAPT [70] | MPM | 22.7 | 5.0 | 90.88 | 90.19 | 85.08 |
| ○ Inter-MAE [29] | MPM | 22.1 | 4.8 | 88.70 | 89.60 | 85.40 |
| ○ Point-M2AE [65] | MPM | 12.9 | 7.9 | 91.22 | 88.81 | 86.43 |
| ○ ACT [8] | MPM | 22.1 | 4.8 | 93.29 | 91.91 | 88.21 |
| ○ PointGPT-B [5] | GPT | 120.5 | 36.2 | 93.60 | 92.50 | **89.60** |
| ○ PointMamba [27] | MPM | 12.3 | - | 93.29 | 91.91 | 88.17 |
| • Point-BERT [60] | MPM | 22.1 | 4.5 | 92.48 | 91.60 | 87.91 |
| • MaskPoint [28] | MPM | 22.1 | 4.5 | 92.17 | 91.69 | 87.65 |
| • Point-MAE [37] | MPM | 22.1 | 4.8 | 92.67 | 92.08 | 88.27 |
| • Point-M2AE [65] | MPM | 12.9 | 7.9 | 93.12 | 91.22 | 88.06 |
| • ACT [8] | MPM | 22.1 | 4.8 | 92.08 | 91.70 | 87.52 |
| • Point-BERT **w/ LCM** | MPM | 3.1 (↓ 86%) | 2.5 (↓ 44%) | 93.55 (↑ 1.07) | 92.43 (↑ 0.83) | 88.57 (↑ 0.66) |
| • MaskPoint **w/ LCM** | MPM | 3.1 (↓ 86%) | 2.5 (↓ 44%) | 93.31 (↑ 1.14) | 91.98 (↑ 0.29) | 87.75 (↑ 0.10) |
| • Point-MAE **w/ LCM** | MPM | 2.7 (↓ 88%) | **1.3** (↓ 73%) | **94.51** (↑ 1.84) | 92.75 (↑ 0.67) | 88.87 (↑ 0.60) |
| • Point-M2AE **w/ LCM** | MPM | **2.5** (↓ 81%) | 6.7 (↓ 15%) | 93.83 (↑ 0.71) | 92.41 (↑ 1.19) | 88.38 (↑ 0.32) |
| • ACT **w/ LCM** | MPM | 3.1 (↓ 86%) | 2.8 (↓ 42%) | 94.13 (↑ 2.05) | **92.66** (↑ 0.96) | 88.57 (↑ 1.05) |

#### 4.2.1 Object Classification

We initially assess the overall classification accuracy of our pre-trained models on both real-scanned (ScanObjectNN [44]) and synthetic (ModelNet40 [57]) datasets. ScanObjectNN is a prevalent dataset consisting of approximately 15,000 real-world scanned point cloud samples from 15 categories. These objects represent indoor scenes and are often characterized by cluttered backgrounds and occlusions caused by other objects. For the ScanObjectNN dataset, we sample 2048 points for each instance and report results without voting mechanisms. We applied simple scaling and rotation data augmentation of previous work [8, 37] in the downstream setting of ScanObjectNN. We reported the results of different models under our downstream setting, with • marking the results. For the ModelNet40 dataset, due to space limitation, we will further analyze its results in Section 5.4.

To ensure a fair comparison, we conducted our experiments following the standard practices in the field (as used in previous work [8, 28, 37, 60, 65]). For each point cloud classification experiment, we used eight different random seeds (0-7) to ensure the robustness and reliability of our results. The performance reported in Table 1 represents the **average accuracy** achieved across these eight trials for each model configuration.

As presented in Table 1, our model has many exciting results. 1) **Lighter**, **faster**, and **more powerful**. When trained from scratch using supervised learning only, our LCM model demonstrates performance improvements of 0.82%, 0.15%, and 1.10% across three variant datasets compared to the Transformer architecture. Similarly, after pre-training (*e.g.*, Point-MAE), our model outperformed the standard Transformer by 1.84%, 0.67%, and 0.60% across the three variants of the ScanObjectNN dataset. Notably, these improvements are achieved despite an **88%** reduction in parameters and a **73%** reduction in FLOPs. This improvement is exciting as it indicates that our architecture is better suited for point cloud data compared to the standard Transformer. Additionally, due to its extremely high

Table 2: Object detection results on ScanNetV2. We adopt the average precision with 3D IoU thresholds of 0.25 ($AP_{25}$) and 0.5 ($AP_{50}$) for the evaluation metrics. † is our reproduction results, due to the lack of detection code in their paper.

| Methods | Pretrain | $AP_{25}$ | $AP_{50}$ |
|---|---|---|---|
| *Supervised Learning Only* | | | |
| VoteNet [38] | ✗ | 58.6 | 33.5 |
| 3DETR [34](*baseline*) | ✗ | 62.1 | 37.9 |
| Transformer [46] | ✗ | 60.5 | 40.6 |
| LCM (**Ours**) | ✗ | 63.8 (↑ 3.3) | 46.4 (↑ 5.8) |
| *Self-Supervised Learning* | | | |
| PointContrast [58] | CL | 58.5 | 38.0 |
| STRL [25] | CL | - | 38.4 |
| Point-BERT [60] | MPM | 61.0 | 38.3 |
| PiMAE [4] | MPM | 62.6 | 39.4 |
| Point-MAE† [37] | MPM | 59.5 | 41.2 |
| Point-M2AE† [65] | MPM | 60.0 | 41.4 |
| ACT [8] | MPM | 63.8 | 42.1 |
| DepthContrast [69] | CL | 64.0 | 42.9 |
| MaskPoint [28] | MPM | 64.2 | 42.1 |
| Point-BERT [60] **w/ LCM** | MPM | **65.3** (↑ 4.3) | **47.3** (↑ 9.0) |
| Point-MAE [37] **w/ LCM** | MPM | 64.7 (↑ 5.2) | 47.2 (↑ 6.0) |
| Point-M2AE [65] **w/ LCM** | MPM | 63.5 (↑ 3.5) | 44.0 (↑ 2.6) |
| ACT [8] **w/ LCM** | MPM | 65.0 (↑ 1.2) | 45.8 (↑ 3.7) |
| MaskPoint [28] **w/ LCM** | MPM | 65.3 (↑ 1.1) | 46.3 (↑ 4.2) |

Table 3: Part segmentation results on the ShapeNetPart. The mean IoU across all categories, i.e., $\text{mIoU}_c$ (%), and the mean IoU across all instances, i.e., $\text{mIoU}_I$ (%) are reported.

| Methods | Pretrain | $\text{mIoU}_c$ | $\text{mIoU}_I$ |
|---|---|---|---|
| *Supervised Learning Only* | | | |
| PointNet++ [40] | ✗ | 81.9 | 85.1 |
| DGCNN [50] | ✗ | 82.3 | 85.2 |
| Transformer [46] | ✗ | 83.9 | 86.0 |
| LCM (**Ours**) | ✗ | 84.6 (↑ 0.7) | 86.3 (↑ 0.3) |
| *Self-Supervised Learning* | | | |
| Transformer-OcCo [48] | CL | 83.4 | 85.1 |
| PointContrast [58] | CL | - | 85.1 |
| CrossPoint [1] | CL | - | 85.5 |
| Point-BERT [60] | MPM | 84.1 | 85.6 |
| IDPT [64] | MPM | 83.8 | 85.9 |
| MaskPoint [28] | MPM | 84.4 | 86.0 |
| Point-MAE [37] | MPM | 84.2 | 86.1 |
| ACT [8] | MPM | 84.7 | 86.1 |
| PointGPT-S [5] | MPM | 84.1 | 86.2 |
| PointGPT-B [5] | MPM | 84.5 | 86.4 |
| Point-M2AE [65] | MPM | 84.9 | 86.5 |
| Point-BERT [60] **w/ LCM** | MPM | 85.0 (↑ 0.9) | 86.5 (↑ 0.9) |
| MaskPoint [28] **w/ LCM** | MPM | 85.1 (↑ 0.7) | 86.6 (↑ 0.6) |
| Point-MAE [37] **w/ LCM** | MPM | **85.1** (↑ 0.9) | 86.6 (↑ 0.5) |
| Point-M2AE [65] **w/ LCM** | MPM | 85.0 (↑ 0.1) | 86.5 (-) |
| ACT [8] **w/ LCM** | MPM | 85.0 (↑ 0.3) | **86.7** (↑ 0.6) |

efficiency, it provides strong support for the practical deployment of these pre-trained models. 2) **Universal**. We have replaced the original Transformer architecture with our LCM model in five different MPM-based pre-training methods. All experimental results are exciting as our model achieved universal performance improvements with fewer parameters and computations, highlighting the versatility of our model. In the future, we will further adapt to additional pre-training methods.

### 4.2.2 Object Detection

We further assess the object detection performance of our pre-trained model on the more challenging scene-level point cloud dataset, ScanNetV2 [6], to evaluate our model's scene understanding capabilities. Following the previous pre-training work [8, 28], we use 3DETR [34] as the baseline and only replace the Transformer-based encoder of 3DETR with our pre-trained compact encoder. Subsequently, the entire model is fine-tuned for object detection. In contrast to previous approaches [4, 8, 28], which necessitate pre-train on large-scale scene-level point clouds like Scan-Net, our approach directly utilizes models pre-trained on ShapeNet. This further emphasizes the generalizability of our pre-trained models.

Table 2 showcases our experimental results, our compact model has shown significant improvements in scene-level point cloud data, such as Point-MAE [37] achieving a 5.2% improvement in $AP_{25}$ and a 6.0% improvement in $AP_{50}$ compared to the Transformer. This improvement is remarkable, and we believe this is primarily due to the presence of a large number of background and noise points in the scene-level point cloud. Using a local constraint modeling approach effectively filters out unimportant background and noise, allowing the model to focus more on meaningful points.

### 4.2.3 Part Segmentation

We also assess the performance of LCM in part segmentation using the ShapeNetPart dataset [3], comprising 16,881 samples across 16 categories. We utilize the same segmentation setting after the pre-trained encoder as in previous works [37, 65] for fair comparison. As shown in Table 3, our LCM-based model also exhibits a clear boost compared to Transformer-based models. These results demonstrate that our model exhibits superior performance in tasks such as part segmentation, which demands a more fine-grained understanding of point clouds.

Table 4: Effects of the Network Structure of the Locally Constrained Compact Encoder.

| Local | Local | MLPs | FFN | Param(M) | ScanObjectNN |
|-------|-------|------|-----|----------|--------------|
| A | ✗ | ✗ | ✔ | 1.4 | 85.45 |
| B | ✗ | ✔ | ✔ | 2.3 | 85.74 |
| C | ✔ | ✔ | ✗ | 1.8 | 87.77 |
| D | ✔ | ✔ | ✔ | 2.7 | 88.06 |

Table 5: Effects of Locally Constrained Mamba-based Decoder.

| Decoder | ScanObjectNN |
|---------|--------------|
| Transformer | 88.76 |
| LAL | 88.38 |
| Mamba | 88.62 |
| Transformer w/ LCFFN | 88.79 |
| LAL w/ LCFFN | 88.51 |
| **Mamba w/ LCFFN** | 89.35 |

## 4.3 Ablation Study

**Effects of Locally Constrained Compact Encoder.** We explore the performance of our locally constrained compact encoder by comparing it with a Transformer-based encoder in classification, detection, and part segmentation. The results from Tables 1, Table 2, and Table 3, obtained solely through supervised learning from scratch, clearly demonstrate the advantages of our LCM encoder over the Transformer-based encoder in terms of performance and efficiency, particularly in detection tasks, with an improvement of up to 6.0% in the $AP_{50}$ metric.

This substantial improvement is attributed to the compact encoder's focused attention on the most crucial information for each point patch, such as local neighborhoods while disregarding unimportant details. This is similar to a redundancy-reducing compression concept, which is crucial for point cloud analysis, especially in large-scale scene-level point clouds where significant redundancy and noise points often exist. Our local constraint approach enables the model to focus on critical areas, leading to a combined improvement in efficiency and performance. Moreover, this redundancy-reducing concept helps our model avoid overfitting the training dataset. We provide detailed explanations of this phenomenon in the Section 5.5.

**Effects of the Network Structure of the Locally Constrained Encoder.** As shown in Figure 5(a), each layer of our locally constrained compact encoder consists primarily of three parts: a locally constrained unit based on k-NN, MLPs mapping unit composed of Down MLP and Up MLP, and the final FFN layer. We explore the effects of each unit separately. Specifically, we train Encoders with different structures from scratch on the ScanObjectNN [44] dataset and test their classification performance. As shown in Table 4, comparing A and B reveals that a simple two-layer MLP without local aggregation does not substantially improve the network's performance. In contrast, the results of C and D compared to A and B demonstrate a significant performance improvement. This improvement is mainly attributed to the introduction of local geometric perception and aggregation. Comparing the results of C and D, the introduction of FFN brings a slight improvement. Therefore, FFN is not indispensable in our compact encoder, but we choose to incorporate FFN to further perform mapping. These experiments further indicate the necessity of local geometric perception and aggregation for point cloud feature extraction.

**Effects of Locally Constrained Mamba-based Decoder.** We further compared the impact of different decoder designs during the pre-training phase. Specifically, we compared the vanilla Transformer-based decoder, our LAL-based decoder, and the vanilla Mamba-based architecture, as well as their performance after incorporating LCFFN. As shown in Table 5, the results indicate that the Vanilla Transformer slightly outperforms the Vanilla Mamba in terms of performance, likely due to the limitation imposed by the simple geometric sequential input sequences on the Vanilla Mamba's capabilities. After incorporating LCFFN, the Mamba decoder exhibits a significant improvement due to the introduction of implicit geometric order. In contrast, the Transformer's improvement is slight because the geometric order is already implicitly captured by self-attention.

## 4.4 Limitation

Our current model does have limitations in handling dynamic importance perception and long-range dependency modeling. Our design prioritizes efficiency, which can be at odds with the increased complexity required for capturing dynamic importance and long-range dependencies. This focus on efficiency led us to simplify the model in certain aspects, and as a result, we did not fully integrate mechanisms for dynamic importance perception and long-range dependency modeling in this version

of our model. Despite these constraints, the current model has demonstrated significant improvements in performance across various tasks. Nevertheless, we also acknowledge that incorporating dynamic importance perception and long-range dependency modeling could further enhance the model's capabilities, particularly in more complex scenarios. We are actively exploring methods to address these limitations in future work.

## 4.5   Conclusion

In this paper, we propose a compact point cloud model, LCM, specifically designed for masked point modeling pre-training, aiming to achieve an elegant balance between performance and efficiency. Based on the idea of redundancy reduction, we propose focusing on the most relevant point patches ignoring unimportant parts in the encoder, and introducing a local aggregation layer to replace the vanilla self-attention. Considering the varying information density between masked and unmasked patches in the decoder inputs of MPM, we introduce a locally constrained Mamba-base decoder to ensure linear complexity while maximizing the perception of point cloud geometry information from unmasked patches. By conducting extensive experiments across various tasks such as classification and detection, we demonstrate that our LCM is a universal model with significant improvements in efficiency and performance compared to traditional Transformer models.

## 4.6   Acknowledgements

This work is supported in part by the National Key Research and Development Project of China (Grant No. 2023YFF0905502), the National Natural Science Foundation of China, under Grant (62302309, 62171248), Shenzhen Science and Technology Program (JCYJ20220818101014030, JCYJ20220818101012025), and the PCNL KEY project (PCL2023AS6-1).

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

# 5  Appendix

## 5.1  An Information Theoretic Perspective of Our Mamba-based Decoder for MPM.

Here, we provide an information-theoretic perspective for our decoder design, using mutual information to qualitatively demonstrate that the Mamba-based SSM can perceive more information from unmasked patches to predict masked patches compared to a Transformer-based self-attention. The mutual information between random variables $X$ and $Y$, $I(X;Y)$, measures the amount of information that can be gained about a random variable $X$ from the knowledge about the other random variable $Y$. Therefore, based on the decoder input's different information densities, we can simply divide the input into $X_1$, representing unmasked patches with higher information density, and $X_2$, representing randomly initialized masked patches with lower information density. As illustrated in Figure 6, after being processed by the decoder, $X_1$ and $X_2$ respectively yield outputs $Y_1$ for unmasked patches and $Y_2$ for masked patches. We reconstruct the masked points based on $Y_2$.

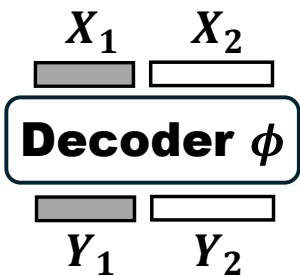

Figure 6: A simple illustration of information processing of MPM decoder.

Ideally, $Y_2$ needs to perceive sufficient geometric priors from both $X_1$ and $X_2$ to recover the masked points, more mutual information represents more recovery potential. Therefore, we would like to maximize the mutual information $I(Y_2; X_1, X_2)$. In what follows, we demonstrate that the mutual information preserved by our proposed Mamba-based decoder is larger than that of the standard transformer decoder.

**Theorem 1.** *Let $Y_1^M, Y_2^M$ and $Y_1^T, Y_2^T$ denote the outputs of the Mamba-based and Transformer-based decoders respectively, $I(Y_2^M; X_1, X_2)$ denote the mutual information preserved by the Mamba-based decoder, and $I(Y_2^T; X_1, X_2)$ denote that of the Transformer-based decoder. We have $I(Y_2^M; X_1, X_2) \geq I(Y_2^T; X_1, X_2)$.*

*Proof.* The first step is to formalize the input-output relation of the two decoding structures. For the Mamba decoder, as defined in [13, 16], the output can be expressed as:

$$Y_2^M = C\bar{A}\bar{B}X_1 + C\bar{B}X_2$$
$$= AX_1 + BX_2.$$

For the Transformer decoder, the attention mechanism can be expressed in the following matrix form:

$$\begin{bmatrix} X_1^\top W_q^\top W_k X_1 & X_1^\top W_q^\top W_k X_2 \\ X_2^\top W_q^\top W_k X_1 & X_2^\top W_q^\top W_k X_2 \end{bmatrix} \begin{bmatrix} W_v X_1 \\ W_v X_2 \end{bmatrix} = \begin{bmatrix} Y_1^T \\ Y_2^T \end{bmatrix}.$$

Thus,

$$Y_2^T = X_2^\top W_q^\top W_k X_1 \cdot W_v X_1 + X_2^\top W_q^\top W_k X_2 \cdot W_v X_2.$$

Compared with the linear relation captured by $Y_2^M$, $Y_2^T$ models higher-order interactions of the input variables. So for any given Mamba parameters $A$ and $B$, there exists Transformer parameters $W_k, W_q, W_v$ and a function $g$, such that $Y_2^T = g(Y_2^M)$.

As $Y_2^T$ is a function of $Y_2^M$, $(X_1, X_2) \to Y_2^M \to Y_2^T$ forms a Markov chain. So $(X_1, X_2)$ and $Y_2^T$ are independent when conditioned on $Y_2^M$, i.e., $p(Y_2^T, X_1, X_2 | Y_2^M) = p(Y_2^T | Y_2^M)p(X_1, X_2 | Y_2^M)$. According to the definition of conditional mutual information, this implies

$$I(X_1, X_2; Y_2^T | Y_2^M) = 0.$$

On the other hand, by the chain rule of mutual information we have

$$I(X_1, X_2; Y_2^M, Y_2^T) = I(X_1, X_2; Y_2^M) + I(X_1, X_2; Y_2^T | Y_2^M)$$
$$= I(X_1, X_2; Y_2^T) + I(X_1, X_2; Y_2^M | Y_2^T).$$

Since we already show that $I(X_1, X_2; Y_2^T | Y_2^M) = 0$, and mutual information is non-negative, we have

$$I(X_1, X_2; Y_2^M) = I(X_1, X_2; Y_2^T) + I(X_1, X_2; Y_2^M | Y_2^T) \geq I(X_1, X_2; Y_2^T).$$

$\square$

## 5.2 Additional Related Work

**Deep Network Architecture for Point Cloud.** With the development of deep learning, various deep neural network-based models [7, 10, 12, 13, 23, 46, 52, 53, 68] have become the mainstream approach for 3D point cloud analysis. PointNet [39], a pioneer in point cloud analysis, introduced an MLP-based network to address the disorder of point clouds. Subsequently, PointNet++ [40] further proposed adaptive aggregation of multiscale features on MLPs and incorporated local point sets for effective feature learning. DGCNN [50] introduced the graph convolutional networks dynamically computing local graph neighboring nodes to extract geometric information. PointMLP [32] suggested efficient point cloud representation solely relying on pure residual MLPs. Recently, many Transformer-based models [20, 34, 37, 62], benefiting from attention mechanisms, have achieved notable improvements in point cloud analysis. However, this led to a significant increase in model size, posing considerable challenges for practical applications. PointMamba [27] first attempted to introduce the Mamba architecture based on the state space model to point clouds, but it still has high complexity and parameters. In this paper, we focus on designing more efficient point cloud architectures specific to pre-training models.

**State Space Models.** State Space Models [14–17, 42] (SSMs) originate from classical control theory and have been introduced into deep learning as the backbone of state space transformations. They combine the parallel training capabilities of CNNs with the fast inference characteristics of RNNs, capturing long-range dependencies in sequences while maintaining linear complexity. The Structured State-Space Sequence model [16] (S4) is a pioneer work for the deep state-space model in modeling the long-range dependency. S5 [42] proposed based on S4 and introduces MIMO SSM and efficient parallel scan. GSS [33] leverages the gating structure in the gated attention unit to reduce the dimension of the state space module. Recently, Mamba [13] with efficient hardware design and selective state space, outperforms Transformers [46] in terms of performance and efficiency. Subsequent works [11, 26, 30, 31, 35, 49, 71] have attempted to introduce Mamba into the visual domain, achieving significant improvements. For example, Vision Mamba [71] and VMamba [30] directly apply Mamba to image processing and design corresponding scanning methods tailored for image data. As for point cloud, PointMamba [27] is the first to introduce Mamba into point cloud analysis, traversing the input sequences from the x, y, and z geometric directions. In this paper, we introduce Mamba into the decoder for masked point modeling and discuss its advantages from an information-theoretic perspective. Additionally, we propose a locally constrained feedforward neural network for Mamba block to adaptively exchange information among geometrically adjacent patches based on their implicit geometry.

## 5.3 Implementation Details

**Top-K Attention Settings in Observation.** In Figure 2, we replace the global attention computation of all patch tokens in Self-Attention with top-K attention computation in both feature space and geometric space to demonstrate the significant amount of redundant computation in the vanilla Transformer. Specifically, after computing all global attention, we further compute a mask matrix. We then add negative infinity to the attention values that need to be masked. After that, we calculate the softmax, where the attention values that were set to negative infinity will become 0, ensuring that the sum of the attention values of the unmasked top-K patches equals 1. We compute different top-K values in both feature space and geometric space, and pretrain the corresponding models. Subsequently, we fine-tune these pretrained models on the three variants of ScanObjectNN using the same top-K attention algorithm, evaluating their accuracy on classification tasks. To minimize error, we report the average accuracy over 10 repeated experiments.

**Positional Encodings.** To complement the 3D spatial information, we apply positional encodings to all encoder and decoder layers. As shown in Figure 4, we first use the Encoder Positional Encoding (EPE) to compute the positional encoding $E^p$ for $C_V$, which is then shared across all layers of the encoder. In the decoding stage, we use the Decoder Positional Encoding (DPE) to calculate the positional encoding for the unmasked $C_V$ and the masked patches $C_M$. These positional encodings are concatenated to form decoder positional encodings $T^p$, which is shared across all layers of the decoder. Following previous work [37, 60], we utilize a two-layer MLP to encode its corresponding 3D coordinates $C_V \in \mathbb{R}^{rN \times 3}$ or $C_M \in \mathbb{R}^{(1-r)N \times 3}$ into $d$-channel vectors $E^p \in \mathbb{R}^{rN \times d}$ or $T^p \in \mathbb{R}^{N \times d}$, and element-wisely add them with the token features before feeding into the attention layer.

**Token Embedding.** We follow the approach of previous works [37, 60] and use a simple Point-Net [39] to map the point patches $\boldsymbol{P}_V \in \mathbb{R}^{rN \times K \times 3}$ from coordinate space to feature space $\boldsymbol{E}^s \in \mathbb{R}^{rN \times d}$. For Point-BERT [60], MaskPoint [28], Point-M2AE [65], and ACT [8], we use the exact same embedding structures as described in their original papers. For Point-MAE [37], we further simplify the embedding, using only a two-layer MLP with dimensions 3-128-384 as the embedding, which further reduces the parameters and computational complexity, as shown in Table 1.

**Object Classification.** Due to significant differences in the settings used for downstream fine-tuning tasks of point cloud classification on the ScanObjectNN [44] dataset in previous self-supervised learning methods [8, 28, 37, 60, 65], such as input point quantity, data augmentation, and the input of the classification task head, we conducted extensive experiments to obtain a performance-friendly downstream fine-tuning setting. Furthermore, we re-evaluated most of the previous methods under our setting, while also conducting a fair comparison between our LCM model and the previous Transformer model under our setting. We mark the results of our downstream fine-tuning setting with an "●" in Table 1.

It can be observed that, compared to the results reported in the original paper, the fine-tuning results using our downstream settings have achieved significant performance improvements. For instance, Point-BERT has shown improvements of **5.34%**, **3.62%**, and **4.99%** on the three variants of the ScanObjectNN dataset, respectively. This improvement is surprising, indicating that there is further potential to be explored in earlier self-supervised learning methods such as Point-BERT, Point-MAE, etc.

**Experiments Compute Resources.** Due to the surprisingly lightweight and efficient of our LCM model, we were able to complete the pre-training tasks using just a single 24GB NVIDIA GeForce RTX 3090 GPU. For downstream classification and segmentation tasks, we used a single RTX 3090 GPU for each. For detection tasks, to accelerate training, we utilized four parallel RTX 3090 GPUs.

**3D Object Detection.** We pre-train and fine-tune Point-MAE for 3D object detection both on ScanNetV2 [6]. In our detection experiments on ScanNetV2, we evaluate our model's understanding of scene-level tasks. Specifically, in the downstream detection fine-tuning experiments, we use 3DETR [34] as the baseline model and replace 3DETR's pre-encoder and encoder with our embed layer and compact encoder, respectively, while keeping all other training settings identical to 3DETR. Unlike many previous methods [8, 28, 65] that require retraining models on ScanNet, we initialize the embed layer and compact encoder with models pre-trained directly on ShapeNet [3]. While this may result in some loss of performance due to the gap between ShapeNet and ScanNet data, it demonstrates the universality of our pre-trained models.

## 5.4 Additional Experiments

**Object Classification on ModelNet40.** ModelNet40 [57] is a well-known synthetic point cloud dataset, comprising 12,311 meticulously crafted 3D CAD models distributed across 40 categories. Following previous work [37, 60, 65], for the ModelNet40 dataset, we sample 1024 points for each instance and report overall accuracy with voting mechanisms. In ModelNet40, we no longer differentiate between the results reported in the paper and our results, as we use the exact same downstream fine-tuning settings as previous methods [8, 28, 37, 65]. Table 6 presents our experimental results, and the overall conclusions are consistent with Section 4.2.1. Our LCM model outperforms the Transformer architecture in terms of both efficiency and performance, indicating the superiority of our model.

**Effects of Locally Constrained K Value.** We further explore the impact of using different numbers of neighbors K in local constraints on performance and efficiency. K=1 indicates no consideration of neighboring information. As K increases, the consideration of local geometry for each point patch also increases, but so does the computational complexity. We train object classification from scratch on the PB-RS-T50 variant of ScanObjectNN, and Figure 7 presents our ablation results. The area of the circle represents the computational floating-point operations (FLOPs). We found that a smaller K, such as 5, is sufficient to achieve satisfactory results in terms of performance and efficiency. Performance initially increases slowly, but when K exceeds a certain threshold, it tends to decline. This is mainly due to larger K values introducing excessive redundancy, thereby limiting the learning capacity.

Table 6: Classification accuracy on synthetic (ModelNet40) point clouds. In ModelNet40, following previous work, we report the overall accuracy (%) with voting mechanisms. For a fair comparison, we used the same downstream task settings as in previous studies.

| Method | #Params(M) | FLOPs(G) | ModelNet40 |
|---|---|---|---|
| *Supervised Learning Only* | | | |
| PointNet [39] | 3.5 | 0.5 | 89.2 |
| PointNet++ [40] | 1.5 | 1.7 | 90.7 |
| DGCNN [50] | 1.8 | 2.4 | 92.9 |
| PointMLP [32] | 12.6 | 31.4 | 94.5 |
| P2P-HorNet [51] | 195.8 | 34.6 | 94.0 |
| Transformer [27] | 22.1 | 4.8 | 92.3 |
| PointMamba [27] | 12.3 | - | 92.4 |
| Transformer [27] | 22.1 | 2.4 | 92.3 |
| **LCM** (Ours) | 2.7 | 0.6 | 93.6 |
| *Self-Supervised Learning* | | | |
| Point-BERT [60] | 22.1 | 2.3 | 93.2 |
| CrossNet [56] | 1.8 | 2.4 | 93.4 |
| Inter-MAE [29] | 22.1 | 2.3 | 93.6 |
| MaskPoint [28] | 22.1 | 2.3 | 93.8 |
| Point-MAE [37] | 22.1 | 2.4 | 93.8 |
| Point-M2AE [65] | 12.8 | 4.7 | 94.0 |
| ACT [8] | 22.1 | 2.4 | 93.7 |
| PointGPT-S [5] | 29.2 | 2.3 | 94.0 |
| PointGPT-B [5] | 120.5 | 18.1 | **94.2** |
| PointMamba [27] | 12.3 | - | 93.6 |
| Point-BERT **w/ LCM** | 3.1 | 1.3 | 93.8 |
| MaskPoint **w/ LCM** | 3.1 | 1.3 | 94.1 |
| PointM2AE **w/ LCM** | 2.5 | 1.7 | 94.1 |
| ACT **w/ LCM** | 3.1 | 1.4 | 93.9 |
| Point-MAE **w/ LCM** | 2.7 | 0.6 | **94.2** |

Figure 7: Effects of locally constrained K value.

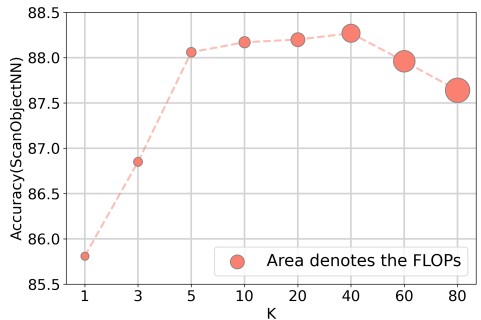

Table 7: Effects of K-NN Space.

| K | Feature K-NN | Geometry K-NN |
|---|---|---|
| 1 | 85.70 | 85.81 |
| 5 | 87.65 | 88.06 |
| 10 | 87.51 | 88.17 |
| 20 | 87.20 | 88.20 |

**Effects of K-NN Space.** We further explored the impact of performing K-NN based on Euclidean distance in both the feature space and the geometric space of our compact encoder. Geometric K-NN in the geometric space imposes explicit geometric constraints, serving as a static importance measure that greatly benefits point cloud analysis. Searching for K-NN based on feature Euclidean distance in the feature space can be considered a simple form of dynamic importance. We analyzed the effect of this approach on point cloud classification from scratch on ScanObjectNN, evaluating geometric K-NN and feature K-NN at different K values.

As shown in Table 7, we found that feature K-NN performed consistently lower than geometric K-NN in almost all cases. This result suggests that the naive idea of assigning dynamic importance to point patches based on Euclidean distance in the feature space does not lead to substantial improvements. Efficient computation of dynamic importance for each point patch remains an area for further exploration.

**Effects of Patch Order and LCFFN for Mamba-based Decoder.** The ordering of input patches significantly impacts our Mamba-based Decoder. To more effectively illustrate this effect on Mamba's SSM model, we analyze the issue from a different perspective. Specifically, we use our Mamba-

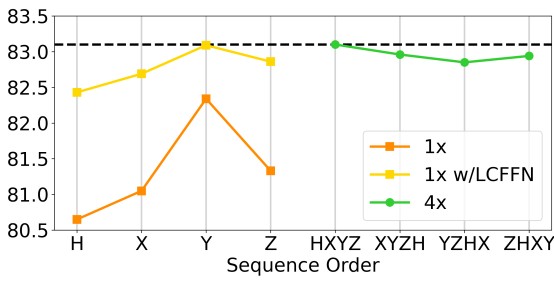
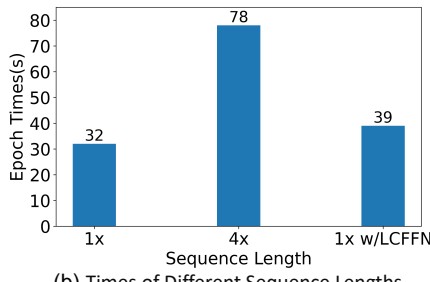

(a) The Effect of Sequence Order on Mamba Model

(b) Times of Different Sequence Lengths

Figure 8: Training and testing curves for different encoders trained from scratch. We present the training and testing curves for both the classification task on ScanObjectNN and the detection task on ScanNetV2. All encoders were not pretrained.

based Decoder as an Encoder to directly extract features from the input point cloud and perform classification on ScanObjectNN. This substitution is straightforward, as our Mamba-based Decoder can also be viewed as an Encoder.

We trained our Mamba-based encoder from scratch for the classification task on the PB-RS-T50 variant of ScanObjectNN without using any data augmentation strategies, and we took the average of ten repeated experiments as the final result. We first experimented with a naive Mamba-based decoder using a traditional FFN to illustrate the impact of different sequence orders on the original Mamba. We selected four different patch ordering methods: sorting by the center point of the patch along the x-axis (X), y-axis (Y), and z-axis (Z), and Hilbert curve [24] ordering (H), as shown by the orange curve in Figure 8 (a). Furthermore, we also conducted experiments with combinations sequences, combining these four orderings "H+X+Y+Z (HXYZ)", "X+Y+Z+H (XYZH)", "Y+Z+H+X (YZHX)", and "Z+H+X+Y (ZHXY)", as shown by the green curve in Figure 8 (a). Finally, based on the single-order sequence, we used our proposed LCFFN to demonstrate the performance of Mamba with added implicit geometric constraints, as shown by the yellow curve in Figure 8 (a). The experimental results, as illustrated in Figure 8, lead us to the following conclusions:

1) *The performance of the Mamba model is greatly influenced by the different orders of input patches.* The orange line represents the results for individual sequences, highlighting that different sequences have a significant impact on the final model performance. For example, the Y-order achieves the highest classification accuracy at 82.34%, while the Hilbert order performs the worst at 80.65%, resulting in a difference of 1.69%.

2) *The more combinations of sequences, the better the representation of point cloud geometry, resulting in improved performance, but also increased computational complexity.* The green line represents the combinations sequences. While different combinations sequences do affect the final model performance, the impact is relatively minor. This indicates that the Mamba model can compensate for information across different sequences, allowing it to capture nearly complete geometric information for each patch. Consequently, this significantly enhances the model's performance. However, this approach leads to a significant increase in computational complexity due to the increase in the length of the input sequence, as shown in Figure 8 (b). The processing time for the sequences of the four orders is approximately $3\times$ longer than that of a single order.

3) *Introducing LCFFN allows for better perception of point cloud geometry through implicit local geometric constraints, thereby mitigating the dependence on sequence order.* The yellow line represents the experimental results of using LCFFN to replace FFN for single-order input. It can be observed that the overall classification accuracy is significantly improved, surpassing the combinations sequence in the y-order and showing only slight differences from the combinations sequence in other orders. Moreover, in terms of runtime efficiency, as shown in Figure 8 (b), our single-order + LCFFN method exhibits a considerable improvement compared to the combinations sequence, indicating the superiority of our design.

## 5.5 Additional Visualization

**Effects of the Compact Encoder from the Perspective of Overfitting.** While our compact encoder has fewer parameters compared to Transformer-based encoders, its performance surpasses that of Transformer-based encoders [46], as analyzed in Section 4.2.1. One significant reason for this lies in the reduced risk of overfitting in downstream tasks due to the redundancy reduction. Given the challenging nature of acquiring point cloud data, existing point cloud datasets for downstream tasks are often small, such as ScanObjectNN [44] and ModelNet40 [57], each comprising just over 10,000 point clouds, and ScanNetV2 [6] with only 1,000 scenes. These dataset sizes are much smaller than those commonly found in image and language tasks. Therefore, fine-tuning in these size-limited datasets can be more prone to overfitting when considerable redundancy exists in the computation.

We visualize the training and testing curves for different encoders on the classification task in ScanObjectNN and the detection task in ScanNetV2 in Figures 9 and 10. Figure 9 illustrates the classification and detection curves for our compact encoder and a Transformer-based encoder after pretraining. It can be observed that during training, the classification accuracy and AP25 metric of the Transformer-based encoder are significantly higher than those of our compact encoder. However, during testing, our compact encoder exhibits superior performance compared to the Transformer encoder. This starkly indicates that the Transformer-based encoder tends to overfit the training set, demonstrating poorer generalization. Conversely, our compact encoder displays stronger generalization capabilities, indicating the superiority of the design of our compact encoder.

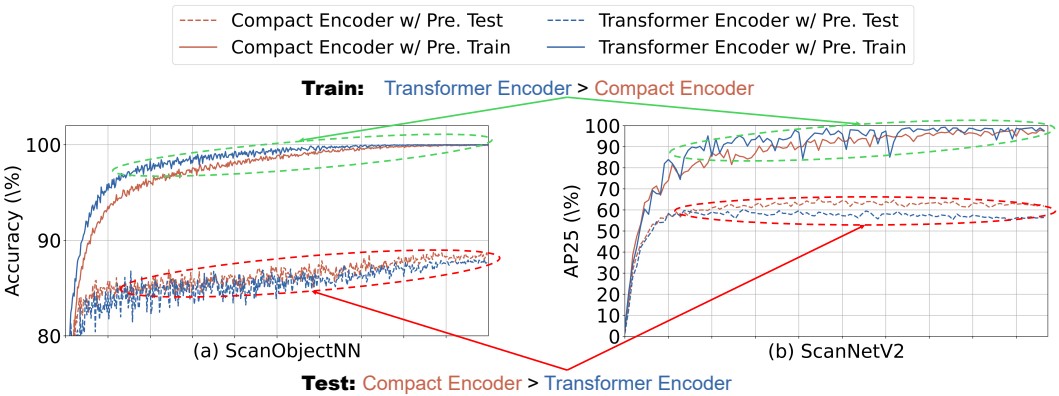

Figure 9: Training and testing curves of different pre-trained encoders. We present the training and testing curves for both the classification task on ScanObjectNN and the detection task on ScanNetV2. All encoders are pre-trained.

Figure 10 displays the classification and detection curves of our compact encoder and the Transformer-based encoder trained from scratch. In comparison to its counterpart in Figure 9, although it shows a slower convergence, the overfitting issue of the Transformer-based encoder still emerges in the late stages of training, reaffirming our conclusion. Meanwhile, the phenomenon of slow convergence in Figure 10 is reasonable as it is an encoder trained from scratch without a better initialization.

**t-SNE Visualization.** We further used t-SNE [45] to visualize the feature distributions extracted by our LCM model and the Transformer. In Figure 11, we visualized the two-dimensional (2D) feature distributions of the two models, pretrained using Point-MAE, when directly transferred to the test set of the ModelNet40 [57] dataset without downstream fine-tuning. In Figure 12, we visualized the 2D feature distributions of the two pre-trained models after fine-tuning on the most challenging variant of the ScanObjectNN [44] dataset, PB-RS-T50, using its test set.

In the 2D t-SNE visualizations, instances from the same category tend to be distributed in relatively clear and tight clusters. The compactness of the feature distributions of different instances from the same category can be viewed as the model's ability to represent features of the same category. A more compact distribution indicates a stronger modeling capability. As shown in Figure 11 and Figure 12, our LCM model achieves more compact feature distributions for instances of the same

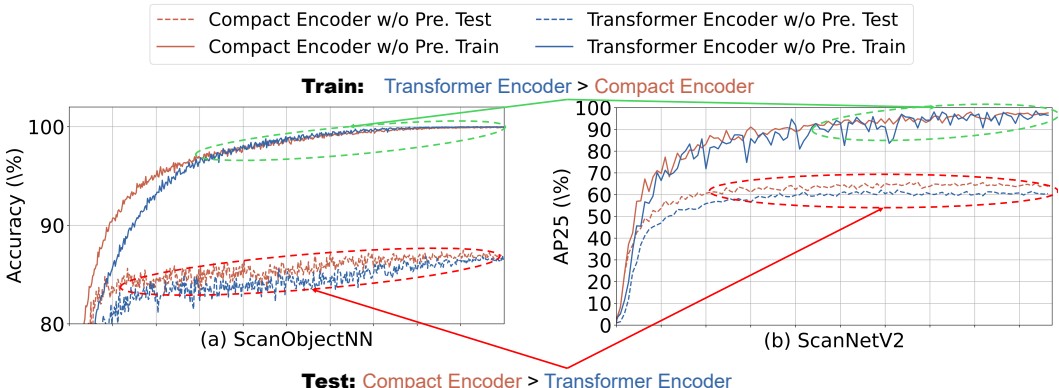

Figure 10: Training and testing curves for different encoders trained from scratch. We present the training and testing curves for both the classification task on ScanObjectNN and the detection task on ScanNetV2. All encoders were not pretrained.

category compared to the Transformer model in most cases, indicating that our LCM model has a stronger ability to model the general representations of the same category.

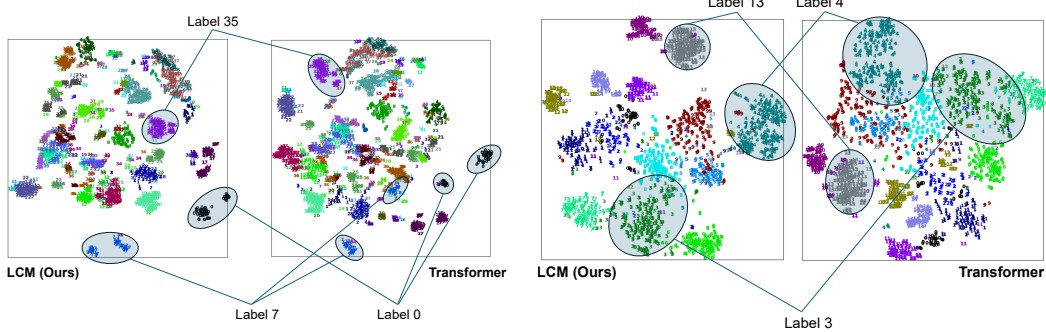

Figure 11: The feature distribution visualization of the pre-trained models on the test set of ModelNet40.

Figure 12: The feature distribution visualization of the fine-tuned models on the test set of ScanObjectNN.

## 5.6 Broader Impacts and Safeguards

Our designed compact point cloud network will greatly facilitates the deployment of existing point cloud pre-training models on resource-constrained devices, which would significantly advance existing point cloud applications. However, the proliferation of more point cloud applications may lead to privacy data leaks, such as personal housing layout point cloud leaks, and human feature point cloud leaks. Therefore, we advocate for the implementation of strict security measures during the actual deployment of applications to prevent malicious access or tampering of data. Below are some corresponding measures:

1) Access Control: Implement stringent access control policies to restrict data access to authorized users or systems only.

2) Data Encryption: Utilize robust encryption algorithms to encrypt sensitive point cloud data during transmission and storage, ensuring its security against unauthorized access.

3) Anonymization: Anonymize sensitive information whenever possible to reduce the risk of data leaks. For instance, remove or blur identifiable information, retaining only essential data for analysis.

