# OpenReview forum: "LCM: Locally Constrained Compact Point Cloud Model for Masked Point Modeling"
_NeurIPS.cc/2024/Conference — NeurIPS 2024 poster_

### Official Review · Reviewer_y4yX · 2024-06-12

**Soundness:** 3
**Presentation:** 2
**Contribution:** 4
**Rating:** 8
**Confidence:** 5

**Summary:**

This paper proposes a locally constrained compact point cloud model (LCM), which consists of a locally constrained compact encoder and a locally constrained decoder based on Mamba. The encoder replaces the self-attention layer with a local aggregation layer, thus achieving a perfect balance between performance and efficiency. The locally constrained Mamba-based decoder is introduced considering the different information densities between masked and unmasked patches in the MPM decoder input. Extensive experimental results show that the LCM greatly outperforms existing Transformer-based models in terms of both performance and efficiency.

**Strengths:**

The novelty of this paper is evident in proposing a locally constrained compact point cloud model by revisiting the shortcomings of the commonly used Transformer-based point mask modeling approach. By acting on five classical point cloud self-supervised methods, it revolutionizes the point cloud self-supervised learning technique in terms of methodology and results.

The theoretical analysis and experimental results in this paper are quite detailed. By localizing and spatially statefulizing the structure of common encoder-decoder architectures, it allows existing self-supervised methods to achieve significant improvements.

The quality of the presentation in this paper is generally excellent, but some of the illustrations and content still need to be improved for clarity. See “Weaknesses” for more details.

The method proposed in this paper is aimed at self-supervised learning of 3D point clouds and has considerable application value in practical applications.

**Weaknesses:**

1. The quality of some figure illustrations in the paper concerns me; The fonts in Figures 4 and 5 are too small and both are in bold, which does not seem to fit the human eye.

2. The elements in Sections 3.3 and 3.4 appear to be improvements over the existing technology and Equation 4-7 should be interpreted differently.

3. Some parts of this paper conflict and it is recommended that the word “section” be capitalized and standardized.

4. The experimental analysis shows that the decrease in the number of parameters of the LCM is extremely obvious, which should be attributed to the Mamba model. It is suggested that the authors analyze the Transformer and Mamba network parameters in detail.

**Questions:**

The significance of this work seems to me and whether the authors will choose to make it fully open source? I hope this work can contribute to the development of the field.

**Limitations:**

No obvious limitations, but some elements require more detailed explanation.

---

> ### Author Rebuttal · Authors · 2024-08-06
>
> ### **@Q1 - Figure quality and font size.**
>
> Thank you for your feedback regarding the quality of the figure and formatting in our paper. We appreciate your attention to these details, as they are crucial for a clear presentation. We acknowledge that the fonts in Figures 4 and 5 are too small and bold, which may affect readability. In the revised version of the paper, we will adjust the font size and weight to ensure they are appropriately sized and easy to read. We will also review all figures to ensure consistency and clarity in visual presentation.
>
> ### **@Q2 - The elements in Sections 3.3 and 3.4 & Equation 4-7.**
>
> Thank you for your suggestion regarding Sections 3.3 and 3.4. In Sections 3.3 and 3.4, we discuss several improvements and modifications to existing technology, which involve the design and implementation of our Locally Constrained Compact Encoder and Locally Constrained Mamba-based Decoder.
>
>  * Section 3.3: The design of the Locally Constrained Compact Encoder introduces innovations based on redundancy reduction. Equations (4) and (5) describe the forward propagation process of the encoder's features. These equations should be interpreted in the context of the local constraints and feature aggregation methods employed in our model. Specifically, the function $f_i(⋅)$ reflects a unique implementation that leverages local geometric constraints, which distinguishes our approach from the standard Transformer. We will provide a more detailed explanation of these modifications in next versions.
>
>  * Section 3.4: Our Locally Constrained Mamba-based Decoder is a design based on mutual information maximization. Equations (6) and (7) outline the forward propagation process within this decoder. The primary changes we made involve the implementation of the functions  $s_i(⋅)$ and $f_i^l(⋅)$. We will also expand on these modifications in more detail in the next version.
>
> Thank you again for your valuable feedback. We will incorporate these clarifications in the revised manuscript.
>
> ### **@Q3 - Consistency in terminology.**
>
> We apologize for any inconsistencies in terminology, particularly with the use of the word “section.” In the revised manuscript, we will capitalize "Section" consistently and standardize its usage throughout the paper. This will include careful proofreading to ensure uniformity in all headings, labels, and references. We appreciate your careful review and constructive comments, which help us improve the clarity and professionalism of our work. These revisions will be reflected in the next version of the manuscript.
>
> ### **@Q4 - Detailed comparison of network parameters.**
>
> Thank you for your insightful feedback. The significant reduction in the number of parameters observed in the LCM model is indeed a crucial aspect of our design, but it is important to clarify that this reduction is primarily due to the Locally Constrained Compact Encoder rather than the Mamba model.
>
> The Mamba model is employed in our Locally Constrained Mamba-based Decoder and is used specifically during the pretraining phase to enhance the reconstruction process. The parameter reduction achieved by the LCM is mainly attributed to the innovative design of the Locally Constrained Compact Encoder, which replaces the standard self-attention layer in Transformers with a more efficient local neighbor aggregation mechanism and compresses the FFN parameters accordingly.
>
> Based on your suggestion, we conducted a detailed comparison of the parameters between the Transformer Encoder Block and our Locally Constrained Compact Encoder Block. The results are as follows, our LCM Encoder has a high parameter reduction compared to the standard Transformer in both the attention layer and the FFN layer.
>
> ```javascript
> Transformer Block 1.773 M{
>     LayerNormal1: 0.001M
>     Self-Attention 0.590 M {
> 	    Q: 0.147 M
> 	    K: 0.147 M
> 	    V: 0.147 M
> 	    Projection: 0.148 M
>     }
>     LayerNormal2: 0.001M
>     FFN: 1.182 M {
> 	    Linear1: 0.59 M
> 	    Linear2: 0.59 M
>     }
> }
> ```
> ```javascript
> LCM Block 0.188 M{{
>     LayerNormal1: 0.001 M
>     Local Aggregation Layer: 0.113 M {
> 	    Down MLP: 0.074 M
> 	    Up MLP: 0.038 M
>     }
>     LayerNormal2: 0.001 M
>     FFN: 0.074M {
> 	    Linear1: 0.037 M
> 	    Linear2: 0.037 M
>     }
> }
> ```
>
> ### **@Q5 - Open source.**
>
> Of course! We are dedicated to advancing the point cloud field and plan to make our work fully open source. We are currently organizing the code and checkpoints, and we aim to release them within the next few weeks. By making these resources available, we hope to facilitate further research and experimentation, and contribute to the broader academic and practical applications of our methods.

---

> > ### Comment · Reviewer_y4yX · 2024-08-08
> > **Reviewer's Response**
> >
> > Thanks to the authors for their careful responses. All my concerns have been addressed. In return, I will firm up my positive assessment of the work and upgrade the rating.
> >
> > As a side note, it is recommended that the authors add some excellent work on self-supervised point clouds.
> >
> > [1] Wu, et al. Self-supervised intra-modal and cross-modal contrastive learning for point cloud understanding. IEEE TMM 2023.
> >
> > [2] Liu, et al. Inter-modal masked autoencoder for self-supervised learning on point clouds. IEEE TMM 2024.

---

> > > ### Author Response · Authors · 2024-08-08
> > > **Response to Reviewer**
> > >
> > > Thank you very much for your positive feedback. We are delighted that our responses have addressed your concerns.
> > >
> > > We also appreciate your recommendation of the excellent self-supervised learning works for point clouds, CrossNet[1] and Inter-MAE[2]. Both are outstanding cross-modal self-supervised learning methods, utilizing 2D-assisted contrastive learning and masked reconstruction, respectively, to learn generalizable 3D representations. In the next version, we will provide a detailed comparison between these works and ours. Your valuable suggestions have significantly contributed to the improvement of our work, and we sincerely thank you once again!
> > >
> > > [1] Wu, et al. Self-supervised intra-modal and cross-modal contrastive learning for point cloud understanding. IEEE TMM 2023.
> > >
> > > [2] Liu, et al. Inter-modal masked autoencoder for self-supervised learning on point clouds. IEEE TMM 2024.

---

### Official Review · Reviewer_jpku · 2024-07-08

**Soundness:** 3
**Presentation:** 3
**Contribution:** 3
**Rating:** 6
**Confidence:** 4

**Summary:**

This paper first proposes a locally constrained compact encoder, which leverages static local geometric constraints to aggregate the most relevant information for each patch token, achieving an elegant balance between performance and efficiency. Moreover, this paper also proposes a locally constrained Mamba-based decoder for masked point modeling. The authors verify the effectiveness and efficiency of the proposed model on multiple pre-training strategies and downstream tasks.

**Strengths:**

1. The model LCM proposed by the authors is novel and very efficient, achieving leading performance with only 2.7M parameters, which is much lower than existing mainstream models.

2. The authors provide a rational explanation for the model design, i.e., an encoder design based on redundancy reduction and a decoder design based on mutual information maximization.

3. Extensive experiments show the effectiveness of the proposed method.

4. The paper is well-organized. The tables, figures and notations are clear.

**Weaknesses:**

Some experimental comparisons are insufficient, and certain details are not clearly described.
1. I noticed that the authors only compared the results of the proposed method with PointGPT-S (NeurIPS 2023) and PointGPT-B. In fact, PointGPT-L offers more powerful performance. Can the author provide a specific explanation for this?

2. The ShapeNet dataset is relatively small, with a limited number of 3D models. Could the author provide results of the proposed method pre-trained on a larger dataset, such as the unlabeled hybrid dataset used in PointGPT (NeurIPS 2023)? This would have a significant impact on demonstrating the generalization ability of the proposed method.

**Questions:**

See the weaknesses.

**Limitations:**

The authors' description of the limitations of their work is reasonable and effective solutions are given.

---

> ### Author Rebuttal · Authors · 2024-08-06
>
> ### **@Q1 - Comparison with PointGPT-L.**
>
> Thank you for your question. We understand the importance of comparing our method with all relevant benchmarks, including PointGPT-L, to provide a comprehensive evaluation.
>
> PointGPT[1], a point cloud pretraining approach published at NeurIPS 2023, proposed extending the generative pretraining techniques from natural language processing (NLP) to point clouds. By partitioning and sorting point patches, the method feeds point embeddings into a transformer decoder for autoregressive prediction. Furthermore, a dual masking strategy was introduced to enhance the learned representations.
>
> The primary reason we initially compared our LCM model only with PointGPT-S and PointGPT-B, and not with PointGPT-L, is due to a significant difference in model size and computational complexity. PointGPT-L has a substantially larger number of parameters compared to our LCM model, which focuses on achieving high efficiency. Specifically, PointGPT-L's parameter count is significantly higher, leading to greater computational load and resource requirements. In contrast, our LCM model is designed to be lightweight and efficient, with a parameter count that is only a fraction of PointGPT-L's.
>
> To address this, we have now included a comparison of our LCM model with PointGPT-L. As shown in the table below, while PointGPT-L does achieve higher accuracy, our LCM model provides a highly efficient alternative, using significantly fewer parameters and computational resources. Specifically, our model's parameter count is only 0.75% of PointGPT-L's, and the computational load is just 1.75% of theirs. This comparison highlights the trade-offs between performance and efficiency, with our LCM model achieving approximately a 100-fold improvement in efficiency.
>
> We hope this explanation clarifies our initial decision and provides a comprehensive understanding of the trade-offs involved.
>
> |   | #Params(M)  | FLOPs(G)  | OBJ-BG  | OBJ-ONLY  | PB-T50-RS  |
> | :------------ | :------------ | :------------ | :------------ | :------------ | :------------ |
> | PointGPT-L  | 360.5  |  74.2 |  95.7 | 94.1  | 91.1  |
> | LCM  | 2.7  | 1.3  |  95.2 | 93.1  |  89.4 |
>
>
> ### **@Q2 - Results on the unlabeled hybrid dataset (UHD).**
>
> Thank you for your question and suggestion. We acknowledge that the ShapeNet dataset is relatively small and may not fully demonstrate the generalization capabilities of our proposed method.
>
> To address this concern, we have conducted additional experiments by pre-training our model on a larger dataset, specifically the unlabeled hybrid dataset used in PointGPT[1]. This dataset is significantly larger and more diverse, providing a more robust evaluation of the generalization abilities of our method.
>
> The results from these additional experiments are presented in the table below. The findings show that our proposed method maintains strong performance and generalization capabilities even when pre-trained on a larger and more diverse dataset. The results underscore the effectiveness of our approach in leveraging large-scale data to learn comprehensive and robust representations.
>
> We believe these additional experiments will provide a clearer understanding of our method's potential and its applicability to a wider range of 3D models and datasets. Thank you for your valuable feedback, which has helped us to strengthen our evaluation.
>
> |   | #Params(M)  | OBJ-ONLY  | PB-T50-RS  |
> | :------------ | :------------ | :------------ | :------------ |
> | ShapeNet55 |  95.18 | 93.12  | 89.35  |
> | UHD  |  95.53 | 93.63  |  90.04 |
>
> [1] Chen, et al. "Pointgpt: Auto-regressively generative pre-training from point clouds." NeurIPS, 2023.

---

> > ### Comment · Reviewer_jpku · 2024-08-10
> > **After the rebuttal**
> >
> > Thank the authors for your rebuttals carefully. My concerns are well addressed. Thus, I keep my original rate to accept this paper.

---

### Official Review · Reviewer_L8Jt · 2024-07-10

**Soundness:** 4
**Presentation:** 3
**Contribution:** 4
**Rating:** 6
**Confidence:** 5

**Summary:**

To address the issues of quadratic complexity and constrained decoders in existing masked point modeling methods based on Transformers, this paper proposes a locally constrained compact point cloud model. First, to tackle the complexity problem, the paper presents an observational experiment with top-K attention to demonstrate the importance of redundancy reduction in point cloud analysis. Based on this redundancy reduction idea, the paper then introduces a locally constrained compact encoder by replacing the self-attention layer with a local aggregation layer. Finally, to overcome the limited reconstruction potential of the original Transformer decoder, the paper proposes a locally constrained Mamba decoder and demonstrates its superiority through both experimental results and information theory analysis.

**Strengths:**

1)	The motivation of this paper is strong, and the mentioned efficiency and reconstruction potential issues of Transformer is important in point cloud analysis;
2)	The paper presents a locally constrained compact point cloud model (LCM) in the field of point cloud self-supervised learning. This model design is quite novel.
3)	The proposed model is universal and achieves impressive results, reaching state-of-the-art effects with a minimal number of parameters.
4)	The paper also offers many well-reasoned and insightful explanations for observed phenomena.

**Weaknesses:**

1)	In the model design, as shown in Figure 5, I noticed that the proposed method seems to perform KNN at each layer. In fact, KNN is a computationally intensive operation. Considering practical applications, this operation could significantly increase the actual inference time of the model. I hope the author can provide a reasonable explanation or experimental validation to address this concern.
2)	The results in Table 1 appear to show the highest performance. However, due to the influence of random seeds, this is not sufficient to reflect the true effectiveness of the model. The authors need to explain how these experimental results were selected, and also report the average performance over multiple trials.
3)	There is an error in the citations. Both the Transformer and PointMamba references in Table 1 under "supervised learning only" point to the paper "Pointmamba: A simple state space model for point cloud analysis."

**Questions:**

My concerns and suggestions have already been outlined in the weaknesses section. I hope the author can provide further explanations on these issues.

**Limitations:**

The author adequately explains the limitations of their work. This static importance of the model may somewhat restrict its actual performance. I hope this limitation can be addressed in future work.

---

> ### Author Rebuttal · Authors · 2024-08-06
>
> ### **@Q1 - The computational cost of KNN.**
>
> Thank you for your question and for highlighting an important aspect of our model's design.
>
> In fact, the computational cost of KNN in LCM is very low. While Figure 5 may give the impression that KNN is performed at each layer, our method actually requires computing the KNN only once at the beginning, not at each layer. This design choice significantly reduces the computational cost associated with KNN operations. Specifically, our low cost is due to the following factors:
>
>  * **Single KNN Computation:** In our approach, the geometric neighbor properties of each point are computed once at the start of the model using KNN. These properties, encapsulated in the graph indices, are then fixed and remain consistent across all layers of the model. This means that after the initial KNN computation, the graph indices are reused in all subsequent layers, eliminating the need for repeated KNN calculations. By sharing the precomputed graph indices across layers, we maintain the enforcement of local constraints while minimizing computational overhead.
>
>  * **Low Cost of Single KNN Computation:** Although KNN has a time complexity of O(n²), the impact on practical applications is mitigated because we only compute it once. Moreover, in common point cloud classification tasks, the number of point patches (n) is typically quite small, ranging from 64 to 128. This relatively small value of n, coupled with a minimal number of nearest neighbors (K), which we set to 5, ensures that the computational cost remains low.
>
> we conducted experiments on the PB-T50-RS variant using the ScanObjectNN dataset. We compared inference time and classification accuracy under different conditions: using shared random indices (without KNN computation), using shared KNN indices (**Ours**), and using individual KNN indices for each layer. As shown in the table below, compared to the random aggregation method without KNN, using shared KNN indices results in only a 0.005 millisecond increase in inference time while improving accuracy by 1.88%. Additionally, compared to performing independent KNN computations at each layer, our method achieves the same level of accuracy with a 0.015 millisecond improvement in speed.
>
> |   | Inference Times  | ScanObjectNN  |
> | :------------ | :------------ | :------------ |
> | w/ Random K Indices (w/o KNN)  | 0.641 ms  | 87.47  |
> | w/ Shared KNN (Ours)  | 0.646 ms  | 89.35  |
> | w/ Individual KNN  | 0.661 ms  | 89.35  |
>
>
> ### **@Q2 - Average performance.**
>
> Thank you for your insightful question. We appreciate your concern regarding the influence of random seeds on the reported performance.
>
> To ensure a fair comparison, we conducted our experiments following the standard practices in the field (as used in Point-BERT[1], MaskPoint[2], Point-MAE[3], Point-M2AE[4], ACT[5]). Specifically, for each point cloud classification experiment, we used eight different random seeds (0-7) to ensure the robustness and reliability of our results. The performance reported in Table 1 represents the highest accuracy achieved across these eight trials for each model configuration.
>
> However, to provide a more comprehensive evaluation of the models' true effectiveness, we also calculated the average performance over these eight trials. This average, which accounts for variability due to random seed selection, offers a more reliable assessment of the models' general performance.
>
> As shown in the three tables below, we report the average classification accuracy of each model on the ScanObjectNN dataset. This mean value, averaged across the eight different random seed runs, demonstrates the stability and consistency of our method. Our LCM model not only achieves the highest performance but also shows superior average performance, consistently outperforming existing transformer-based methods. This analysis confirms the true effectiveness and reliability of our approach.
>
> **(1) ScanObjectNN（OBJ-BG）**
> |   | Point-BERT[1]  | MaskPoint[2]  | Point-MAE[3]  | Point-M2AE[4]  | ACT[5]  |
> | :------------ | :------------ | :------------ | :------------ | :------------ | :------------ |
> | Transformer  |  92.48 |  92.17 |  92.67 | 93.12  | 92.08  |
> | **LCM (Ours)** | 93.55  | 93.31  | 94.51  | 93.83  | 94.13  |
>
> **(2) ScanObjectNN（OBJ-ONLY）**
> |   | Point-BERT[1]  | MaskPoint[2]  | Point-MAE[3]  | Point-M2AE[4]  | ACT[5]  |
> | :------------ | :------------ | :------------ | :------------ | :------------ | :------------ |
> | Transformer  |  91.60 |  91.69 |  92.08 | 91.22  | 91.70  |
> | **LCM (Ours)**| 92.43  | 91.98  | 92.75  | 92.41  | 92.66  |
>
> **(3) ScanObjectNN（PB-T50-RS）**
> |   | Point-BERT[1]  | MaskPoint[2]  | Point-MAE[3]  | Point-M2AE[4]  | ACT[5]  |
> | :------------ | :------------ | :------------ | :------------ | :------------ | :------------ |
> | Transformer  |  87.91 |  87.65 |  88.27 | 88.06  | 87.52  |
> | **LCM (Ours)** | 88.57  | 87.75  | 88.87  | 88.38  | 88.57  |
>
> ### **@Q3 - Citation error.**
>
> Thank you for your thorough review. We will address this issue in the next version.
>
> [1] Yu, et al. "Point-bert: Pre-training 3d point cloud transformers with masked point modeling." CVPR, 2022.
>
> [2] Liu, et al. "Masked discrimination for self-supervised learning on point clouds." ECCV, 2022.
>
> [3] Pang, et al. “Masked autoencoders for point cloud self-supervised learning.” ECCV, 2022.
>
> [4] Zhang, et al. “Point-m2ae: Multi-scale masked autoencoders for hierarchical point cloud pre-training.” NeurIPS, 2022.
>
> [5] Dong, et al. “Autoencoders as cross-modal teachers: Can pretrained 2d image transformers help 3d representation learning?” ICLR, 2023.

---

> > ### Comment · Reviewer_L8Jt · 2024-08-09
> >
> > Thank you for the authors' response. I have carefully reviewed the reply, and the authors have addressed my concerns to a satisfactory extent. I am inclined to accept this paper and look forward to seeing the corresponding revisions in the final version.

---

### Official Review · Reviewer_UfZW · 2024-07-15

**Soundness:** 2
**Presentation:** 3
**Contribution:** 3
**Rating:** 6
**Confidence:** 5

**Summary:**

This paper proposes LCM, a locally constrained compact point cloud model, to improve the efficiency and performance of point cloud processing tasks. It consists of a locally constrained compact encoder and a locally constrained Mamba-based decoder. A locally constrained compact encoder utilizes the proposed local aggregation layer to replace self-attention to achieve an elegant balance between performance and efficiency. A locally constrained Mamba-based decoder introduces LCFFN after each Mamba SSM layer, maximizing the perceived point cloud geometry from unmasked patches. By focusing on local geometric constraints and leveraging SSM, the model achieves a balance between performance and efficiency.

**Strengths:**

1. Introducing a locally constrained compact encoder and Mamba-based decoder is a creative solution that improves both performance and efficiency.

2.  The paper offers a thorough explanation of the proposed model, including the local aggregation layers and the integration of state space models, making the methodology clear and reproducible.

3. The paper provides comprehensive experimental results demonstrating the effectiveness of the proposed method across multiple tasks and datasets.

**Weaknesses:**

1. ***[Static Importance Perception]*** The reliance on static local constraints may limit the model’s ability to dynamically identify and focus on important regions of the point cloud, potentially missing critical information.

2. ***[Long-Range Dependency Modeling]*** While the local constraints improve efficiency, they might not capture long-range dependencies as effectively as self-attention mechanisms, which could be a limitation in some applications.

3. ***[Scene-level semantic segmentation]*** It would be better if the author could provide fine-tuning performance on semantic segmentation with scene-level point cloud datasets like ScanNet or ScanNet++ to make the claim stronger.

**Questions:**

1.The authors have validated the effectiveness of the proposed LCM model in scene data through indoor scene detection tasks. However, is the proposed model also effective for other scene tasks, such as indoor semantic segmentation?

2.In the supplementary materials, the authors used the proposed mamba-based decoder as an encoder to verify the impact of patch order and LCFFN. I have the following two questions:

1) Firstly, the performance listed in Figure 8(a) seems to show a significant disparity compared to the classification performance listed in Figure 2(c) and Table 1. What is the reason for this disparity?

2) Secondly, does this disparity indicate that the effectiveness of the proposed mamba-based decoder is inferior to that of the Transformer and the proposed LCM encoder?

**Limitations:**

The limitations presented by the authors are reasonable. However, the solutions to these limitations appear to lack specificity. I would like to inquire whether the authors have any concrete solutions regarding "efficient dynamic importance." This is merely to ask if the authors have any reasonable and specific ideas without requiring actual experimental results.

---

> ### Author Rebuttal · Authors · 2024-08-06
>
> ### **@Q1 - Static Importance Perception & Long-Range Dependency Modeling & Limitations.**
>
> Thank you for your insightful question. Our current model does have limitations in handling dynamic importance perception and long-range dependency modeling. Our design prioritizes efficiency, which can be at odds with the increased complexity required for capturing dynamic importance and long-range dependencies. This focus on efficiency led us to simplify the model in certain aspects, and as a result, we did not fully integrate mechanisms for dynamic importance perception and long-range dependency modeling in this version of our model.
>
> Despite these constraints, the current model has demonstrated significant improvements in performance across various tasks. Nevertheless, we also acknowledge that incorporating dynamic importance perception and long-range dependency modeling could further enhance the model's capabilities, particularly in more complex scenarios.
>
> We are actively exploring methods to address these limitations in future work. Specifically, we are investigating the use of approximate nearest neighbor (ANN) algorithms to model dynamic importance and long-range dependencies more efficiently. By leveraging non-exact neighbor queries, we aim to balance the computational cost with the need for more sophisticated modeling, ensuring that we can extend the model's capabilities without compromising on efficiency.
>
> ### **@Q2 - The effectiveness of LCM in scene-level semantic segmentation.**
>
> To demonstrate the generalization capability of our proposed LCM model on scene-level data, we further validated its effectiveness in the scene-level semantic segmentation task. Specifically, we replaced the Transformer encoder used in Point Transformer V3 [1] with our LCM encoder to create our model for semantic segmentation. We trained our LCM model on the three most commonly used indoor scene point cloud datasets: ScanNet [2], ScanNet200 [3], and S3DIS [4], and reported their segmentation results on the validation sets. In all three datasets, we report the mean Intersection over Union (mIoU) percentages and benchmark these results against previous backbones.
>
> As shown in the table below, our LCM model continues to perform well in semantic segmentation tasks across multiple scene datasets. Notably, in the ScanNet200 dataset, it outperforms the state-of-the-art Point Transformer V3 by 0.9. In the other two datasets, its performance is comparable to the state-of-the-art Point Transformer V3. These results collectively demonstrate the strong generalization capability of our model on scene-level point cloud data.
>
> | Methods  | ScanNet  | ScanNet200  | S3DIS  |
> | :------------ | :------------ | :------------ | :------------ |
> | PointNeXt [5]  | 71.5  | -  | 70.5  |
> | OctFormer [6]  | 75.7  | 32.6  | -  |
> | Point Transformer V1 [7]  | 70.6  | 27.8  | 70.4  |
> | Point Transformer V2 [8]  | 75.4  | 30.2  | 71.6  |
> | Point Transformer V3 [1]  | 77.5  | 35.2  | 73.4  |
> | LCM (Ours)  | 77.6  | 36.1  | 73.4  |
>
> ###  **@Q3 - mamba-based decoder.**
>
> **1.Explanation of the Performance Difference.**
>
> The observed difference primarily arises from the use of different data augmentation strategies during fine-tuning on downstream tasks. In Figure 8(a), all experimental results reflect the classification accuracy on the ScanObjectNN dataset when trained from scratch. Without any data augmentation, our Mamba encoder combined with LCFFN and y-order sorting achieved an accuracy of 83.1%. In contrast, our LCM encoder, trained from scratch using scaling and rotation data augmentation strategies, reached an accuracy of 86.3%. This significant discrepancy is largely attributable to the different data augmentation methods employed. To ensure a fairer comparison and minimize the impact of data augmentation on performance, we further evaluated the networks' performance without using any data augmentation strategies in the table below.
>
> **2.Comparison of the Effectiveness of the Mamba-based Decoder with Transformer and LCM Encoder**
>
> We further compared the performance of three different encoder structures—standard Transformer, our Mamba+LCFFN, and LCM encoder—on downstream tasks without using any data augmentation. As shown in the table, our LCM encoder and LCM Mamba architecture as encoders significantly outperformed the standard Transformer architecture, demonstrating the effectiveness of these two architectures. Moreover, when comparing the LCM encoder with the LCM Mamba architecture, the LCM encoder proved to be more efficient and effective, largely due to our redundancy reduction strategy. Overall, both the LCM encoder and LCM Mamba architectures are highly efficient network architectures; however, the LCM encoder is better suited for encoder roles, while the LCM Mamba architecture is better suited for decoder roles.
>
> | Methods  | #Params. (M)  | OBJ-BG | OBJ-ONLY  | PB-T50-RS  |
> | :------------ | :------------ | :------------ | :------------ | :------------ |
> | Transformer  | 22.1  | 87.61  | 87.87  | 82.79 |
> | Mamba+LCFFN(y_order)  | 12.7  | 88.79  | 88.98 | 83.09|
> | LCM Encoder  | 2.7  | 89.85  |89.11 | 83.38 |
>
> [1] Wu, et al. “Point Transformer V3: Simpler Faster Stronger.” CVPR, 2024.
>
> [2] Dai, et al. “ScanNet: Richly-annotated 3d reconstructions of indoor scenes.” CVPR, 2017.
>
> [3] Rozenberszki, et al. “Language-grounded indoor 3d semantic segmentation in the wild.” ECCV, 2022.
>
> [4] Armeni, et al. “3d semantic parsing of large-scale indoor spaces.” CVPR, 2016.
>
> [5] Qian, et al. “Pointnext: Revisiting pointnet++ with improved training and scaling strategies.” NeurIPS, 2022.
>
> [6] Peng-Shuai Wang. “Octformer: Octree-based transformers for 3D point clouds.” TOG, 2023.
>
> [7] Zhao, et al. “Point transformer.” ICCV, 2021.
>
> [8] Wu, et al. “Point transformer v2: Grouped vector attention and partition-based pooling.” NeurIPS, 2022.

---

> > ### Comment · Reviewer_UfZW · 2024-08-11
> >
> > Thank you for addressing my questions. I believe that our discussion on the weaknesses of LCM is valuable, and I look forward to seeing more comprehensive solutions in the future. I remain positive about this paper and increase my score to wa.

---

### Author Rebuttal · Authors · 2024-08-04

We sincerely thank all the reviewers for their time and their thoughtful comments and questions. We are pleased to find that:

* All reviewers unanimously appreciated the novelty and effectiveness of our work, recognizing it as a creative solution that revolutionizes the point cloud self-supervised learning technique in terms of both methodology and results.
* All reviewers commended the thorough theoretical and experimental explanations we provided for the proposed method.
* All reviewers acknowledge the significant improvements and promising performance achieved by our approach.
* Reviewer y4yX recognizes the practical application value of the LCM. Designing deployable point cloud algorithms has been our constant pursuit.

We carefully considered all questions, concerns, and comments provided by reviewers. We attempted our best to address the questions as time allowed. We believe the comments & revisions have made the paper stronger and thank all the reviewers for their help.  We provide detailed responses to each review separately, please find individual responses to your questions below.

---

### Comment · Area_Chair_YWSX · 2024-08-10
**Discussions**

Dear Reviewers,

Thank you for your efforts. Please review the rebuttals, engage in the discussions, and provide your final ratings.

Thank you again for your valuable contributions.

AC

---

### Decision · Program_Chairs · 2024-09-25

**Decision:**

Accept (poster)

**Comment:**

This paper presents a novel approach called the Locally Constrained Compact Point Cloud Model (LCM), which features a locally constrained compact encoder and a Mamba-based decoder. By replacing the traditional self-attention mechanism with a local aggregation layer, the proposed model achieves an effective balance between performance and efficiency. The LCM method is supported by comprehensive experiments and rigorous comparisons with state-of-the-art models. All reviewers recommend accepting the paper, and the Area Chair finds no compelling reasons to overturn their recommendations, thus recommending its acceptance.